

**Heterogeneous Phototransformation of Halogenated Polycyclic Aromatic**
**Hydrocarbons: Influencing Factors, Mechanisms and Products**
Yueyao Yang[1, 2, 3], Yahui Liu[1, 2, 3], Guohua Zhu[2], Bingcheng Lin[2], Shanshan Zhang[1, 2, 3], Xin Li[1, 2, 3],
Fangxi Xu[4], He Niu[4], Rong Jin[2, *], Minghui Zheng[1, 2, 3]
[1] State Key Laboratory of Environmental Chemistry and Ecotoxicology, Research Center for Eco-
Environmental Sciences, Chinese Academy of Sciences, Beijing, 100085, China
[2] School of Environment, Hangzhou Institute for Advanced Study, University of Chinese Academy of
Sciences, Hangzhou, 310024, China
[3] College of Resource and Environment, University of Chinese Academy of Sciences, Beijing, 100049,
China
[4] Zhejiang Taizhou Ecological and Environmental Monitoring Center, Taizhou 318000, China
* Correspondence to: jinrong@ucas.ac.cn (R. Jin)



## ABSTRACT

Chlorinated and brominated polycyclic aromatic hydrocarbons (XPAHs) are emerging pollutants widely found in atmospheric particulate matter (PM). However, their environmental transformation mechanisms remain poorly understood. In this study, we collected PM samples of varying sizes over a year for XPAH analysis and found the average concentrations of XPAHs peaked in winter and were dominated by $PM_1$ (47.0%). Correlation analysis with relevant meteorological parameters showed strong associations between XPAH fluctuations and PM, temperatures, and humidity. Hence, controlled laboratory experiments were conducted to explore the influence of particle size, sunlight duration, temperature, humidity, and oxidant concentrations on XPAH transformation. Our results indicated that the transformation rates of XPAHs were influenced by the parent polycyclic aromatic hydrocarbon structures, with phenanthrene < fluoranthene < pyrene < benz[a]anthracene ≈ anthracene < benzo[a]pyrene, as well as the substitution of halogens: chlorinated < brominated. Furthermore, the photo irradiation promoted the heterogeneous transformation of XPAHs, with this process being accelerated by the increased concentrations of reactive oxygen species and elevated temperature, peaking at the humidity level of 45%. The transformation products were identified by nontarget analysis. We then proposed phototransformation pathways for XPAHs, suggesting a mechanism involving dechlorination followed by oxidation. Predictions were made regarding the persistence, bioaccumulation, long-range transportation, and toxicities of XPAHs and their transformation products, showing a decrement in environmental risks as the transformation progressed. This study provides novel insights into the primary influencing factors for particulate XPAH variations and the mechanisms of heterogeneous phototransformation.

**KEYWORDS:** Photochemistry; Influencing factors; Heterogeneous phototransformation; Transformation mechanism; Transformation pathways;



## 1. Introduction

Chlorinated and brominated polycyclic aromatic hydrocarbons (ClPAHs and BrPAHs; XPAHs) are halogenated derivatives of polycyclic aromatic hydrocarbons (PAHs) that have garnered considerable attention in recent years due to their heightened persistence, toxicity, and bioaccumulation relative to their parent PAHs (Jin et al., 2017b; Ma et al., 2013; Nishimura et al., 2017; Ohura et al., 2013). At present, research on XPAHs primarily encompasses the following aspects: (1) Establishment of pre-treatment and instrumental methods (Jin et al., 2023; Liu et al., 2019b; Noro et al., 2023; Sei et al., 2021; Takikawa et al., 2023), (2) Environmental detection across various environmental media (Jin et al., 2020; Xie et al., 2021), and (3) Identification of anthropogenic sources. XPAHs have been reported to exist in the air (Jin et al., 2017c; Kakimoto et al., 2014; Nilsson and Ostman 1993), soil (Sun et al., 2013; Zhang et al., 2006), water (Shiraishi et al., 1985), sediment (Ohura et al., 2015), and organisms (Jin et al., 2017a; Liu et al., 2019b; Nishimura et al., 2017; Ohura et al., 2015; Xia et al., 2019). The sources, such as industrial thermal processes (Yang et al., 2022b), electronic waste decomposition (Wang et al., 2022), and vehicular emissions (Deng et al., 2023) have been identified, by detection of XPAHs in the stack gas and fly ash emitted from these sources (Jin et al., 2017b; Nishimura et al., 2017; Yang et al., 2022a). However, a significant aspect of research appears to have been overlooked: the environmental transformation. Variations in the transformation mechanisms of XPAH congeners in environmental matrices can result in differences in their environmental fate and associated risks.

The photochemical processes have been verified to represent a marked elimination pathway for atmospheric organic chemical species (Hu et al., 2021; Laskin et al., 2015; Malecha and Nizkorodov 2016). For example, PAHs have been confirmed to undergo oxidation (Zhu et al., 2022), or thorough fragmentation (Hu et al., 2021; Zhang et al., 2024) under atmospheric photochemical reactions. Therefore, despite the absence of direct study, given their structural similarities to PAHs, XPAHs also possess the potential to undergo similar processes. A study by Ohura et al. also confirmed the photochemical transformation of ClPAHs when being exposed to light in cyclohexane solvent (Ohura et al., 2008). Field observations provided additional evidence suggesting that the photochemistry plays a crucial role in atmospheric XPAH transformation. For instance, certain studies have shown that concentrations of ClPAHs in particulate matter (PM) were slightly higher during nighttime than during daytime (Ma et al., 2013; Ohura et al., 2013), indicating that daytime photochemistry contributed to the transformation of XPAHs. For BrPAHs, although no direct nighttime versus daytime concentration





comparison has been conducted, prior research indicated that BrPAHs might exhibit greater instability
under photo irradiation than ClPAHs (Ohura et al., 2009).
Heterogeneous reactions were identified to be the key mechanisms driving the transformation of
atmospheric organic compounds, e.g., PAHs (Jia et al., 2019), during these photochemistry processes
(Zhang et al., 2023). Atmospheric PM acts as a significant carrier for both environmental pollutants and
catalysts, serving as a medium for heterogeneous reactions. These reactions could be influenced by
various environmental factors. For instance, the reactive oxygen species (ROS) have been identified to
accelerate the phototransformation of polychlorinated naphthalene (PCNs), which are two-ring ClPAHs,
on the surface of silica gel (Kang et al., 2021). In addition, the temperature and humidity have been
noted to influence the lifetime of atmospheric organics (Shiraiwa et al., 2011). For example, the
heterogeneous oxidation mechanisms of organophosphate flame retardants were found to be
significantly affected by humidity (Liu et al., 2019a). In the case of particulate XPAHs, heterogeneous
reactions may also play a crucial role in their transformation. However, the influencing factors, specific
mechanisms, pathways, and products remain unclear, necessitating further exploration.
This study aims to unravel the mechanisms, influencing factors, pathways, and products of XPAH
heterogeneous transformation on PM. To achieve this, we conducted field studies complemented by
meticulously designed laboratory experiments and nontarget organic compound analysis. Initially, we
collected year-range particle samples of various sizes along with relevant meteorological data. These
samples were subsequently analyzed for XPAHs. Through multivariate parameter analysis, we explored
XPAH fluctuations correlated with meteorological data to pinpoint key influencing factors. Subsequently,
controlled laboratory experiments were designed and conducted to unveil the heterogeneous
transformation of XPAHs under the influence of particle size, humidity, temperature, and atmospheric
oxidant content. The persistence, bioaccumulation, long-range transportation, and toxicities of the
transformation products were then assessed to determine the environmental risks associated with XPAH
transformation. Therefore, the findings of this study contribute comprehensive insights into the
mechanisms and environmental risks involved in the fate of XPAHs in the environment.
**2. Materials and methods**
**2.1 Experimental materials**
In this study, a comprehensive investigation was conducted on 41 XPAHs with 2 to 5 benzene rings,
along with an analysis of the corresponding parent PAHs. The parent PAHs are abbreviated as follows:





Nap (naphthalene), Phe (phenanthrene), Ant (anthracene), Fluor (fluoranthene), Pyr (pyrene), BaA
(benz[a]anthracene), and BaP (benzo[a]pyrene). The congeners in the ClPAH and BrPAH groups are
described as Cl-, $Cl_2$-, $Cl_3$-, $Cl_4$-, Br-, and $Br_2$-, indicating the presence of monochlorinated, dichlorinated,
trichlorinated, tetrachlorinated, monobrominated, and dibrominated PAH derivatives, respectively, with
the numbers denoting the substituting positions. Specific names, abbreviations, and molecular structures
of 20 ClPAHs and 21 BrPAHs are listed in **Table S1**. As reported in our previous study (Jin et al., 2017c),
standards, isotopically labeled internal standards, and recovery standards for XPAHs and PAHs were
commercially obtained.

**2.2 Sample collection and extraction**

Atmospheric particles of three sizes ($PM_1$, $PM_{2.5}$, $PM_{10}$) were collected in Hangzhou from March
2023 to February 2024 using medium-flow samplers (Wuhan Tianhong Instruments Co., Ltd., China).
These samplers were positioned atop a school building at a height of approximately 15 meters above
ground level. The sampling site (30° 8′ 15″ N, 120° 4′ 17″ E) was situated in the Xihu District of
Hangzhou, with no industrial area within a five-kilometer radius. Samples were collected in monthly
cycles, at a flow rate of 0.1 $m^3$/min. Quartz fiber filters were employed to capture particles. Subsequently,
samples were collected, dried, weighed, and stored at −18 ℃.
Samples were extracted with a mixture of dichloromethane and hexane (1:4, v: v) by an auto-
Soxhlet extractor (Universal Extractor E-800, Buchi, Germany). The extracted samples were then
purified using an active silica column and concentrated to 50 μL using a rotary evaporator and nitrogen
blower. Specific processes for the extraction and clean-up processes can be found in our previous study
(Jin et al., 2017c).

**2.3 Experiments design for heterogeneous transformation of particulate XPAHs**

In this study, we designed a photo-transformation device that provided complete confinement and
precise control over the experimental conditions. The reaction unit employed a xenon lamp (light
intensity: 100 mW·$cm^{-2}$) as the primary light source with an AM1.5 filter, which can achieve a good fit
with the sunlight spectrum, effectively simulating the outdoor solar radiation (Cao et al., 2020; Wang et
al., 2020). The entire photolysis reaction unit comprises a gas supply, mass flowmeters, a dryer (with
molecular sieve and color silica gel), a bubbler containing Milli-Q water, a xenon lamp, an optical
reactor, a quartz reaction vessel, a temperature control system, gas absorption bottles, and a relative
humidity monitoring component (**Fig. 1a**). To accurately emulate atmospheric conditions, model



particles and a composite mixture comprising 41 XPAHs were meticulously prepared. Sequentially, 20
μL of the XPAHs mixture (1 mg/L) or individual congener solution was deposited onto the surface of the
prepared layer of silica particles ($M_{XPAHs}$: $M_{PM}$ = 2μg/g) (**Fig. 1b** and **Fig. S1**). The settings of the
concentrations were based on previous studies on the XPAH concentrations per particle mass (Jin et al.,
2017a; Jin et al., 2018). Specific information on the devices and experiments are described in **Text S1**.
To comprehensively explore the impact of various factors, including particle sizes (100 nm, 2 μm,
10 μm), temperatures (10°C, 20°C, 30°C), humidity (RH=30%, 45%, 60%), oxidant concentrations (0'%
(+ tert-butyl alcohol; TBA), 0%, 1%, 3%, 5%, 10%), and irradiation duration (0 min, 10 min, 30 min, 60
min, 180 min), on the phototransformation mechanism of XPAHs, a series of experiments were
conducted (**Text S1**). The reacted gas was directed into a toluene solution for analysis of XPAHs in the
tail gas, with less than 1% of the XPAHs escaping through volatilization during the reaction time. Dark
control groups were conducted in each experiment to ensure that the difference between the two sets of
experiments was due to the transformation effect caused by photo irradiation. Upon completion of the
reactions, the particle layer within the reaction vessel was sonicated with 200 μL of toluene, followed by
centrifugation of the supernatant. The resulting solution was then transferred into a centrifuge tube for
subsequent product analysis.
**2.4 Instrumental analysis**
Analysis of the XPAHs and PAHs was conducted using gas chromatography coupled with magnetic
sector high-resolution mass spectrometry (HRGC-HRMS, DFS, Thermo Fisher Scientific, USA)
equipped with an electron ionization source. Specific information on the parameters for the oven and MS
can be found in our previous study (Jin et al., 2017a). The analytical program and instrumental
parameters for the analysis of PAHs were set according to "CalEPA Method 429".
Non-target analysis of the transformation products of XPAHs was performed using a Trace 1310
GC coupled to a quadrupole-Orbitrap MS (Thermo Fisher Scientific, USA). Data were collected and
processed using Thermo Scientific TraceFinder 5.1 software. High-resolution mass spectra of unknown
compounds were deconvoluted into pure spectra using the Deconvolution Plugin of TraceFinder
software and then verified against the standard mass spectra from the commercial library NIST 2014.
Specific information on the non-target analysis was shown in **Text S2**.
**2.5 Environmental behavior and toxicity assessment of XPAHs and their transformation products**
To assess the ecological risk and environmental characteristics—specifically, persistence, long-



range transport potential, and bioaccumulation—of transformation products of XPAHs, this study
employed the KOWWIN, KOAWIN, BCFBAF, and Level III fugacity models within EPI Suite 4.1 to
compute various physicochemical properties of compounds. Essential environmental parameters,
including molecular weight, octanol-water partition coefficient (Kow), air-water partition coefficient
(Kaw), and half-lives in air, water, and soil, were subsequently introduced into the Pov-LRTP tool
(Concha and Manzano 2023). Subsequently, the P-B-LRTP score is developed based on persistence
(Pov), characteristic travel distance (CTD), and transfer efficiency (TE) values to prioritize the screened
compounds. The respiratory toxicity model in ProTox 3.0(Banerjee et al., 2024) was used to predict the
toxicity of the products, assessing their median lethal dose (LD50) and toxicity class.
$$P - B - LRTP\ Score_i = LogPov + LogBAF + LogTE$$
**2.6 Quality assurance and quality control**
For the analysis of actual atmospheric PM samples, the recovery rates of internal standards ranged
from 52% to 105%. In laboratory simulation experiments, the recovery rates of XPAHs ranged from
78% to 115%, while those of PAHs ranged from 53% to 120%. These recovery rates met the
requirements for the detection and analysis of persistent organic pollutants. The detection limits (LODs)
ranged from 0.17 to 1.9 fg/m$^3$ for ClPAHs, and from 0.23 to 1.6 fg/m$^3$ for BrPAHs. A blank sample was
analyzed together with each batch of samples, and the relative concentrations of all XPAH congeners
were below the detection limits.

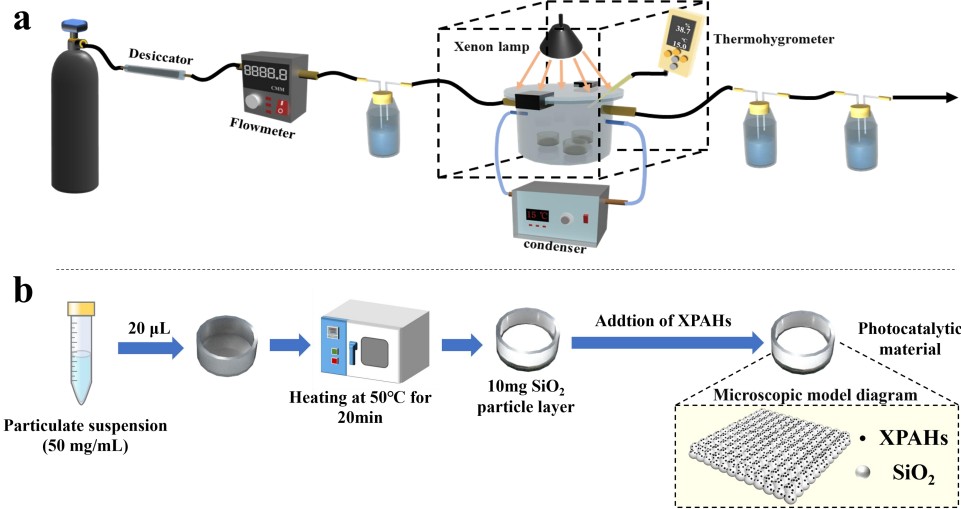


**Fig. 1.** (a) The laboratory photochemical transformation setup. (b) Particulate matter preparation process.



# 3. Results and discussion

## 3.1 Occurrence levels and congener profiles of particulate XPAHs

During the sampling period, PM concentrations ranged from 41.5 to 652.7 $\mu g/m^3$, with an average concentration of $130.0 \pm 167.6$ $\mu g/m^3$ (**Fig. S2a**). $PM_1$ had the highest proportion, with an average of $47.0\% \pm 13.5\%$, while the proportions of $PM_{1-2.5}$ and $PM_{2.5-10}$ were comparable. Notably, the peak concentration was recorded in April 2023. This surge coincided with a dust storm originating from the north, leading to heightened levels of suspended dust in the atmosphere and a significant spike in PM levels. Barring exceptional weather conditions, PM concentrations in other seasons were approximately twice as high as those observed in summer and autumn.

The concentrations of $\sum_{21}ClPAHs$ in the particles ranged from 0.6 to 61.5 $pg/m^3$ (mean: $12.1 \pm 16.9$ $pg/m^3$), while those of $\sum_{18}BrPAHs$ ranged from N.D. to 5.4 $pg/m^3$ (mean: $0.6 \pm 1.5$ $pg/m^3$) during the sampling period (**Fig. 2a**). These values are lower than those reported in previous studies, such as Beijing, China ($\sum_{19}ClPAHs$: $128.8 \pm 102.6$ $pg/m^3$, $\sum_{19}BrPAHs$: $9.5 \pm 13.8$ $pg/m^3$) (Jin et al., 2017a); Ulsan, South Korea ($\sum_{11}BrPAHs$: 1.62 $pg/m^3$) (Vuong et al., 2020); and Shizuoka, Japan ($\sum_{20}ClPAHs$: $133 \pm 53$ $pg/m^3$) (Ohura et al., 2013). For distribution in particles of different sizes, ClPAHs had the highest fraction in $PM_1$ (mean: 45.4%, range: 29.4–65.9%) and comparable proportions in $PM_{1-2.5}$ (mean: 28.1%, range: 13.0–52.0%) and $PM_{2.5-10}$ (mean: 26.5%, range: 8.46–3.5%). Conversely, BrPAHs showed no significant variance across the three particle size ranges, with concentrations of $PM_1$ (mean: 35.9%, range: 12.5–68.2%), $PM_{1-2.5}$ (mean: 35.7%, range: 14.3–55.2%) and $PM_{2.5-10}$ (mean: 28.4%, range: 1.25–59.2%) (**Fig. 2b**). In total, over 70% of XPAHs were bound to particles with diameters smaller than 2.5 $\mu m$.

Concentration trends of XPAH homologue groups were as follows: ClFluor > ClBaP > ClAnt > ClPyr > ClPhe > ClBaA (**Table S2**); BrPyr > BrPhe > BrAnt > BrTriph > BrBaA > BrFluor (N.D.) (**Table S3**). The distribution profiles of XPAH congeners in $PM_{10}$ in Hangzhou are shown in **Fig. S2b**, **Fig. S2c**, and **Table S4**. Among ClPAHs, 2-ClPhe/9-ClPhe, $1,5-Cl_2Ant/9,10-Cl_2Ant$, 1-ClPyr, 3-ClFluor, $3,8-Cl_2Fluor$, and 6-ClBaP were found to be predominant throughout the year. This distributions aligned with findings of prior studies (Jin et al., 2017a; KITAZAWA et al., 2006; Ma et al., 2013). It is worth noting that 6-ClBaP, characterized by the highest molecular weight and highest toxic equivalent factor investigated within ClPAHs investigated in our study, demonstrated the highest concentration proportion. BrPAHs are predominantly constituted by 2-BrPhe, 9-BrPhe, 7-BrBaA, and $1,6-Br_2Byr$. This presented a

notable departure from previous literature which predominantly identified 3-BrFluor, 1,8-Br$_2$Ant, and 1-
BrPyr as the primary congeners of BrPAHs in atmospheric PM in Beijing (Jin et al., 2017a). This
disparity underscores variances in their respective sources or transformations.

**3.2 Key environmental factors influencing temporal variations of particulate XPAHs**

Extreme weather events such as sandstorms and prolonged light rainfall have been excluded in the
following discussion. The average concentrations of PM and BrPAHs reached nadirs during summer and
autumn while showing higher levels in spring and winter (**Fig. S2d**). Conversely, ClPAH concentrations
remained relatively stable during spring and summer, decreased in autumn, and peaked in winter. The
seasonal characteristics of ClPAHs with different parent PAH structures also varied (**Fig. 2c**). Except for
ClBaA, the remaining ClPAHs reached their highest concentrations during winter. ClFluor showed
elevated concentrations in both spring and winter, whereas ClAnt demonstrated higher concentrations in
autumn and winter. ClPhe maintained relatively consistent concentrations across the remaining three
seasons. The seasonal characteristics of ClPAHs and BrPAHs (**Fig. 2d**) also differed. Concentrations of
BrPAHs varied significantly with the seasons, with no congener detected in summer and high
concentrations of BrPhe, BrAnt, and BrPyr in spring and winter, likely influenced by climatic conditions
such as temperature and sunshine. This disparity can be attributed to the ease of generation from sources
and greater atmospheric stability of ClPAHs, while BrPAHs may be subject to influences from
atmospheric processes (Ohura et al., 2009).
Eight meteorological parameters were collected throughout the sampling period: wind speed, wind
direction, air pressure, temperature, humidity, total irradiation, sunshine duration, and rainfall. Details
are listed in **Table S5**, and the results of multifactor correlation analysis are shown in **Fig. 2e**. Pearson
correlation analysis revealed significant positive correlations between ClPAHs and BrPAHs (P < 0.001,
R = 0.96), PM (P < 0.01, R = 0.96), and humidity (P < 0.05, R = 0.76). BrPAHs exhibited a significant
positive correlation with humidity (P < 0.01, R = 0.81). The impact of humidity was notably significant,
as an increase in humidity tended to facilitate the upward adsorption of XPAHs onto PM. However, a
previous study (Vuong et al., 2020) from Ulsan, South Korea has reported a negative correlation
between humidity and XPAHs. This observation suggested that the relationship between XPAHs and
humidity varied across different regions. Additionally, ClPAHs shew a significant negative correlation
with temperature (P < 0.01, R = –0.90), suggesting that higher temperatures corresponded to lower
ClPAH concentrations. This further elucidated the phenomenon of lower ClPAH levels observed during



the summer and autumn seasons. Other meteorological factors didn't show significant correlations with
ClPAHs or BrPAHs (P > 0.05), possibly due to the intricate interplay of multiple factors under natural
conditions. Hence, the specific roles of meteorological conditions such as sunlight intensity, duration of
sunshine, temperature, and humidity warranted further investigation.







**Fig. 2.** (a) The fluctuations of concentrations of PM , $\Sigma_{21}$ClPAHs, and $\Sigma_{19}$BrPAHs in the atmosphere. (b) The

proportions of ClPAHs and BrPAHs across different PM diameters. (c-d) Seasonal distributions of ClPAH (c) and



BrPAH (d) congeners, excluding events of extreme conditions, arranged from outermost to innermost layers:
ClPhe, ClAnt, ClPyr, ClFluor, ClBaA, and ClBaP; BrPhe, BrAnt, BrPyr, BrTriph, and BrBaA. (e) Pearson
correlation analysis between ClPAHs、BrPAHs and ClPAHs/PM with meteorological parameters (wind speed,
wind direction, air pressure, temperature, humidity, total radiation, sunshine duration, and rainfall) under non-
extreme weather conditions.
**3.3 General transformation mechanisms of particulate XPAHs under photo irradiation**
According to both preliminary research(Li et al., 2023) and the experimental results mentioned
above, it is evident that meteorological conditions can significantly impact the concentrations and
distributions of XPAHs on PM. Controlled laboratory experiments were conducted in this study to unveil
the heterogeneous transformation mechanisms of XPAHs.
The transformation ratios of XPAHs were calculated based on the ratios of transformated XPAHs to
the initial concentration (($C_0$-$C_t$)/$C_0$). With the increase of irradiation time, a general trend of
transformation was observed across XPAH congeners (**Fig. 3**). Overall, the transformation rates of
ClPAHs appeared to follow the sequence: ClPhe < ClFluor < ClPyr <ClBaA ≈ ClAnt < ClBaP (**Fig. S3**).
This pattern aligned with the previously reported trends in PAH photolysis rates (Phe < BaA < Ant ≈
BaP) (Zhao et al., 2017), indicating strong influence by the parent PAH structures. In a combined
consideration of ClPAHs and their parent PAHs, shown as "ClPAHs+PAHs" in **Fig. S3**, we also
observed the formation of PAHs and decreases of total concentrations of ClPAHs and PAHs. This
indicated that there were dechlorination of ClPAHs and simultaneous fragmentation of parent PAH
structures during the transformation of ClPAHs.
In addition, the substitution numbers in the structures had a strong influence on the transformation
of ClPAHs: with increment of the substitution numbers, the transformation ratios progressively
decreased (**Fig. 3**). For example, after 1 h of irradiation, transformation ratios were approximately 20%
for 3-ClPhe and 2-ClPhe/9-ClPhe, while 10% for 9,10-$Cl_2$Phe. An exception was observed for ClAnt
congeners (1,4-$Cl_2$Ant or 1,5-$Cl_2$Ant/9,10-$Cl_2$Ant < 1-ClAnt/2-ClAnt or 9-ClAnt < 1,5,9,10-$Cl_4$Ant. This
difference could be attributed to the fact that the investigated low-chlorinated ClAnts are dechlorination
products of high-chlorinated ClAnts. While the substitution position had some impact, it was not as
significant as the number of chlorines. For instance, both 2,5-$Cl_2$Fluor and 3,8-$Cl_2$Fluor demonstrate
comparable photolysis extents, with approximately 20% transformation after 30 min of irradiation, and
transformation ratios of 68.8% and 49.4% after 3 h of irradiation, respectively.



For BrPAHs (**Fig. S4**), the overall transformation ratio ranked as follows: BrPhe < BrAnt < BrPyr
≈ BrFluor < BrBaA < BrBaP. The transformation ratio ranking between BrPAHs and ClPAHs exhibited
disparities, implying that distinct halogen substitutions might yield diverse transformation effects.
BrPAHs degraded more rapidly than ClPAHs. For example, BrPyr degraded by 60% after 30 min of
irradiation, while ClPyr only degraded by less than 40%. Additionally, the increase in bromination
degree didn't appear to have a notable effect on the transformation rate of BrPAHs, which differed from
ClPAHs.

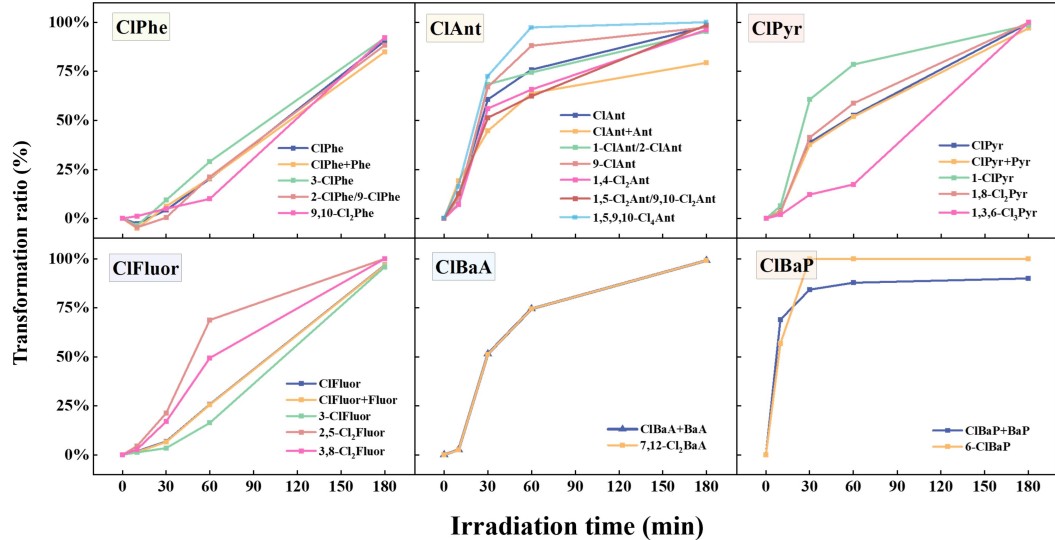


**Fig. 3.** The relationships between transformation ratios of ClPAHs and photo irradiation time.
**3.4 Influencing factors for heterogeneous transformation of particulate XPAHs**
The influencing factors, i.e., particle size, relative humidity, reactive oxygen species content, and
temperature on XPAH transformation have been investigated (**Text S1**). Although previous studies have
revealed variations in XPAH concentrations on particles of different sizes (Jin et al., 2017a; Lara et al.,
2022; Ma et al., 2013), the influence of particle size (100 nm, 2 μm, and 10 μm) on transformation
efficiency was found to be not notably significant in simulated experiments (**Fig. S5 and Text S3**).
The influence of humidity varied among ClPAHs with different parent structures (**Fig. 4a**). For
instance, the transformation of ClPhe and ClFluor slowed down with increasing humidity, whereas the
transformation of ClAnt, ClPyr, ClBaA, and ClBaP increased with higher humidity levels, with the
highest transformation ratio observed at 45% humidity. Additionally, we found that the impact of



humidity on transformation was not consistent for ClPAH congeners in the same homologue groups. For
example, transformation of 9,10-Cl$_2$Phe decreased more sharply than the other ClPhe congeners with the
increment of humidity. The combined influence of photo irradiation and humidity enhanced the
transformation of individual ClPAHs (**Fig. S6**). For example, under 45% humidity, ClPyr exhibited an
average transformation of 50% in darkness, while this rate increased to 75% under photo irradiation.
Possible reasons could be that the addition of photo irradiation drove the formation of OH radicals (·OH),
which could join the breakdown of molecules (Zhang et al., 2023). In contrast, the acceleration of
transformation with increasing humidity was relatively universal for BrPAHs, with the most significant
effects observed within the 30-45% humidity range (**Fig. S6c** and **Fig. S6d**).
Transformation ratios of XPAHs increased as temperatures rose, with the most significant
transformation observed at 30°C (**Fig. 4b**), indicating that the elevated temperature promote the
transformation of XPAHs. The transformation ratio of each congener gradually increased with
temperature compared to dark conditions (**Fig. S7a**). Under photo irradiation at the same temperature,
the transformation ratios of nearly all XPAH congeners increased by more than 50% compared to dark
conditions. This indicated that the breakdown of XPAH molecules was enhanced by the photo irradiation.
In both photo irradiation and dark conditions, the transformation ratios of BrPAHs (**Fig. S7c** and **Fig.**
**S7d**) exceeded those of ClPAHs. This phenomenon can be attributed to the lower bond energy of C-Br
(291 kJ/mol) compared to C-Cl (345 kJ/mol), which is consistent with the findings of previous studies
by Ohura et al (Ohura et al., 2009).
Adding H$_2$O$_2$ to the reaction system simulates the effects of oxidants present in the atmosphere on
the impact of XPAH transformation under photochemical conditions. The transformation of ClPAHs
accelerated with the increase of H$_2$O$_2$ concentration (**Fig. 4c** and **Fig. S8**). TBA was introduced to
eliminate the ·OH effect in the control group. Under 1-hour photo irradiation conditions, the
transformation rate ranking of ClPAHs is as follows: ClBaP > ClBaA > ClPyr > ClPhe > ClAnt >
ClFluor. The transformation of ClFluor showed no significant change with H$_2$O$_2$ content, indicating
relative stability. Additionally, compared to monochlorinated compounds, the overall transformation
ratios of polychlorinated compounds are relatively slower and less influenced by H$_2$O$_2$ (except for
1,5,9,10-Cl$_4$Ant). This also suggested that the dechlorination process was more pronounced in high-
chlorinated compounds, while the transformation of low-chlorinated substances was mainly due to ring-
opening reactions. The situation was similar for BrPAHs, with transformation rates faster than those of



ClPAHs.

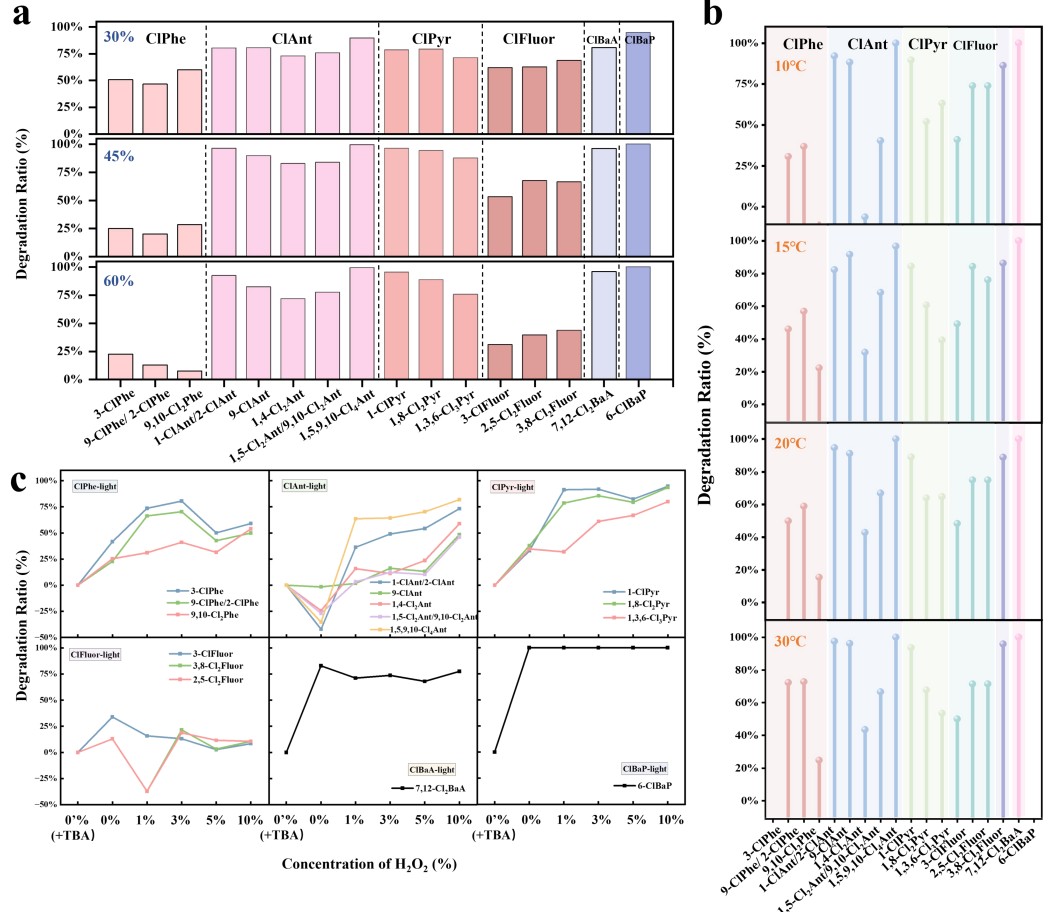


**Fig. 4.** Transformation ratios of ClPAHs under varying (a) humidity levels, (b) temperature conditions and (c)
$H_2O_2$ concentrations, with photo irradiation.
**3.5 Heterogeneous transformation pathways of XPAHs under photo irradiation**

According to the results above, there are dechlorination, direct ring-opening, or ring-opening

induced by oxidation processes involved in the breakdown of XPAHs. To elucidate the specific
transformation pathways of XPAHs, we conducted photolysis experiments on highly chlorinated XPAHs,
including $1,4-Cl_2Ant$, $9,10-Cl_2Phe$, $1,8-Cl_2Pyr$, $2,5-Cl_2Fluor$, $7,12-Cl_2BaA$, and $6-ClBaP$, individually.
Non-target analysis was then employed to recognize the transformation products of these congeners.
Specific mass spectra can be found in **Fig. S9**, and the relative compounds are listed in **Table S6**. The



predominant products were quinones, ketones, hydroxyl-bearing compounds, and ring-opened products,
consistent with previous findings on PAH phototransformation products (Jia et al., 2019; Zhao et al.,
2017). Surprisingly, no chlorinated oxides were detected. Further analysis of the products from
experiments on both ClPAHs and BrPAHs revealed no significant differences. As a result, we confirmed
a hypothesis proposed in previous studies (Ohura et al., 2009) that the transformation of XPAHs
underwent dehalogenation to form PAHs before oxidation.
The transformation pathways are presented in **Fig. 5**. According to the non-target analysis results of
the photolysis products, the primary products for most ClPAHs, with the exception of ClFluor, were the
parent PAHs in the initial step. And for ClFluor, the initial step involved not only the dechlorination, but
also ring-opening, resulting in the generation of 2,7-dichlorofluorene and 9-chlorofluorene. Research
findings suggested that in the atmosphere, the primary reactions for the destruction of aromatic
compounds were the addition or substitution with ·OH (Dang et al., 2014). At the same time, the
substitution of H or Cl in the main structures of ClPAHs or PAHs by the ·OH could influence the
products of the next steps. Therefore, the subsequent steps might include ring opening, oxidation, or
hydrolysis.
Specifically, there were three potential pathways for the transformation of 1,4-Cl$_2$Ant (**Fig. 5a** and
**Fig. S9a**). In the first pathway, 1,4-Cl$_2$Ant could undergo substitution by ·OH to form A2 (1,4-
dihydroxyanthracene, C$_{14}$H$_{10}$O$_2$), which further oxidized to produce A3 (1,4-anthraquinone, C$_{14}$H$_8$O$_2$).
For the second and third pathways, 1,4-Cl$_2$Ant underwent dechlorination to yield A1 (Ant, C$_{14}$H$_{10}$). The
subsequent steps were then similar to those of Ant, with oxidation leading to the formation of A4 (9,10-
Anthracenediol, C$_{14}$H$_{10}$O$_2$), and further oxidation to A5 (9,10-Anthraquinone, C$_{14}$H$_8$O$_2$) as the second
pathway. In the third pathway, ring-opening occurred to form A6 (2-Methylnaphthalene, C$_{11}$H$_{10}$),
followed by subsequent chain-breaking ring-opening to form A7 (Nap, C$_{10}$H$_8$), A8 (1,2,3,4-
Tetrahydronaphthalene, C$_{10}$H$_{12}$), and A9 (2-Methoxyphenol, C$_7$H$_8$O$_2$), ultimately yielding A10 (3-
Nonanol, C$_9$H$_{20}$O) and A11 (3-Nonanone, C$_9$H$_{18}$O). Similar transformation pathways were observed for
9,10-Cl$_2$Phe (**Fig. 5b** and **Fig. S9b**), 1,8-Cl$_2$Pyr (**Fig. 5c** and **Fig. S9c**), and 7,12-Cl$_2$BaA (**Fig. 5e** and **Fig.**
**S9e**), involving dechlorination to generate PAHs, followed by attack by ·OH to produce phenols, further
oxidizing to form quinones, acids, and esters. Additionally, 9,10-Cl$_2$Phe exhibited an additional
oxidation pathway involving ring-opening oxidation, yielding B9 (9-Fluorenone, C$_{13}$H$_8$O), followed by a
series of chain-breaking ring-opening reactions to sequentially generate B10 (Benzophenone, C$_{13}$H$_{10}$O),



B11 (3'-Methylacetophenone, $C_9H_{10}O$), and B12 (3-Nonanone, $C_9H_{18}O$). 1,8-$Cl_2$Pyr also had two
additional potential transformation pathways. In the first pathway, uniform cleavage occurred both above
and below the pyrene molecule, generating C4 (Nap, $C_{10}H_8$) and C5 (2-Methylnaphthalene, $C_{11}H_{10}$). In
the second pathway, diagonal cleavage resulted in the identification of two products: C6 (2-(2,5-
dimethylphenyl)-1,4-dimethylbenzene, $C_{14}H_{14}$) and C7 (4,4'-Dimethyldiphenyl, $C_{14}H_{14}$). C6 underwent
attack by ·$OH/O_2^-$ radicals to ultimately form C8 (Benzophenone, $C_{13}H_{10}O$), followed by further ring-
opening cleavage to produce C9 (3'-Methylacetophenone, $C_9H_{10}O$). Similarly, C10 (2-
Phenylprocpionaldehyde, $C_9H_{10}O$) was formed by oxidation-induced chain-breaking of C8.
For ClFluor (**Fig. 5d** and **Fig. S9d**), the initial step involved ring-opening, generating D1 (2,7-
$Cl_2$Fle, $C_{13}H_8Cl_2$) and D2 (9-ClFle, $C_{13}H_9Cl$). Subsequent dechlorination and ring-opening led to the
formation of D3 (4,4'-Dimethyldiphenyl, $C_{14}H_{14}$), D4 (2-(2,5-dimethylphenyl)-1,4-dimethylbenzene,
$C_{14}H_{14}$), and D5 (Benzophenone, $C_{13}H_{10}O$) in the samples. In the second pathway, the initial
dechlorination process resulted in the formation of D6 (Fluor, $C_{16}H_{10}$), which yielded D7 (1H-Indene,
2,3-dihydro-4,7-dimethyl-, $C_{11}H_{14}$) and D9 (Nap, $C_{10}H_8$). D7 underwent bond cleavage to form D8 (1H-
Indene, octahydro-, $C_9H_{16}$), while D10 (1,2-Diethylbenzene, $C_{10}H_{14}$) was the ring-opening product of D9.
Further oxidation (alcohol to aldehyde) sequentially yielded D11 (1,2-Dicarboxybenzene, $C_8H_6O_4$) and
D12 (Dibutyl phthalate, $C_{16}H_{22}O_4$).
6-ClBaP, as a mono-chlorinated compound (**Fig. 5f** and **Fig. S9f**), exhibited the highest
phototransformation ratio among all ClPAH congeners. After dechlorination, the initial step generated
F1 (BaP, $C_{20}H_{12}$). Among them, positions 3, 6, and 12 of BaP were particularly reactive, and ·OH attacks
led to the formation of F2, F3, and F4 (6-Benzo[a]pyrenol, 6,12-Dihydroxybenzo[a]pyrene, and 9,12-
Dihydroxybenzo[a]pyrene, $C_{20}H_{12}O$), subsequently further generated F5, F6 and F7 (6-Benz[a]pyrenone,
Benzo[a]pyrene-6,12-dione, and Benzo[a]pyrene-3,6-dione, $C_{20}H_{10}O_2$). Under light exposure, F6
underwent further oxidation and ring-opening to produce F8 (Lapachol, $C_{15}H_{14}O_3$), F9 (2,6-
Diisopropylnaphthalene, $C_{16}H_{20}$), and F11 (Dimethyl phthalate, $C_{16}H_{22}O_4$), as reported by S. Zhao et al
(Zhang et al., 2023). The exploration of XPAH transformation is limited by the absence of quantification
of the products. Further studies are necessary to elucidate the specific molecular assignments.





**Fig. 5.** (a-f) The transformation pathways and relative products of ClPAHs. ((a) 1,4-Cl₂Ant; (b) 9,10-Cl₂Phe; (c) 1,8-Cl₂Pyr; (d) 2,5-Cl₂Fluor; (e) 7,12-Cl₂BaA; (f) 6-ClBaP).

**3.6 Assessments of persistence, bioaccumulation, long-range transportation and toxicities of XPAHs and their transformation products**

Studies have suggested that transformation products of organic pollutants might exhibit distinct environmental behaviors and heightened ecological toxicity (Zhang et al., 2023). For the assessed P-B-LRTP scores of XPAHs and their transformation products in this study (**Fig. 6a** and **Table S6**), it could



be observed that as the transformation pathways progressed, the scores of the transformation products
decreased. For instance, the score of 9,10-Cl$_2$Phe was 5.07, which decreased to 2.13 after dechlorination,
and further oxidation products have even lower scores ranging from 1.65 to -1.27 (9,10-
Difluorenquinone to 1,2-Benzenedicarboxylic acid). However, there were also transformation products
with relatively high scores, such as B9 (9H-Fluorene-9-one, score: 1.57) and B10 (Benzophenone, score:
1.81), which warrant special attention in future studies.

To further investigate the toxicities of these transformation products, the lethal doses (LD50) and

toxicity levels (**Fig. 6b** and **Table S7**) were prediectd by the respiratory toxicity model in ProTox 3.0
(Banerjee et al., 2024). The findings indicated that as the transformation pathways progressed, the LD50
values of the products generally increased, except for 2,5-Cl$_2$Fluor, indicating a general toxicity
decrement alongside XPAH transformation. For instance, the LD50 of 9,10-Cl$_2$Phe was 886 mg/kg with
a toxicity level of 4. Among its transformation products, only B1 (Phe) had an LD50 (316 mg/kg, level 4)
lower than that of 9,10-Cl$_2$Phe, while the toxicities of other oxidation and ring-opening products were
lower. However, previous studies on the aryl hydrocarbon receptor activities of XPAHs in yeast assays
reported the opposite results on toxicities of 9,10-Cl$_2$Phe and Phe: the relative equivalent potency of
9,10-Cl$_2$Phe was found to be much higher than Phe. In the case of 2,5-Cl$_2$Fluor, its inherently high LD50
(4220 mg/kg, level 5) resulted in most of its products having higher toxicity levels compared to the
parent compound. Overall, most transformation products have toxicity levels lower than their precursors.
However, given the disparities between model predictions and experimental results, further toxicity
experiments are needed to substantiate the changes in toxicity during the transformation process of
XPAHs.

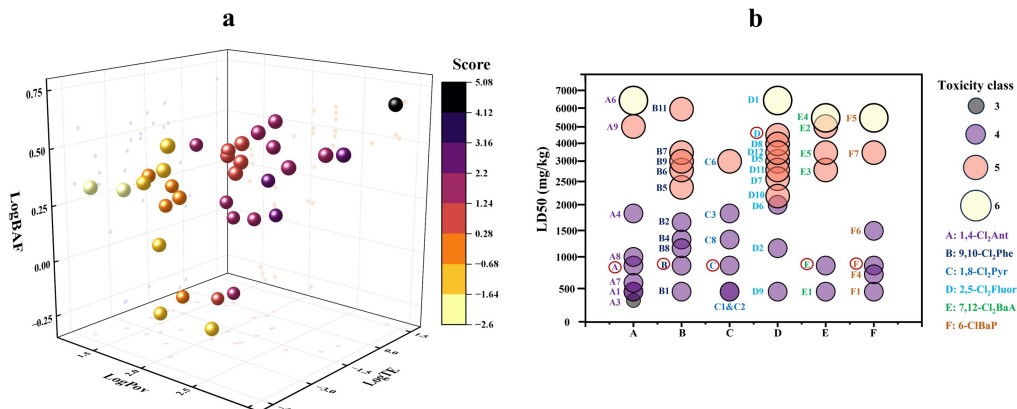




**Fig. 6.** (a) Prediction of environmental behaviors of XPAHs and their transformation products. (b) Respiratory
toxicities (LD50) of XPAHs and their transformation products predicted by ProTox 3.0 model.
**4.   Conclusion**
In summary, this study elucidated the mechanisms, influencing factors, pathways, and products
involved in the conversion of XPAHs on PM through comprehensive field sampling and laboratory
simulations. Experimental findings revealed that the molecular structures of PAHs exerted a significant
influence on the conversion process, with dehalogenation and cleavage of the parent ring structure being
prominent features of XPAHs conversion. The number and position of substituents further modulated the
conversion dynamics. Key environmental parameters, including humidity, temperature, and $H_2O_2$
concentration, were identified as critical factors impacting conversion efficiency. The resulting
conversion products and pathways were systematically hypothesized and confirmed, indicating a
progressive decrease in environmental risks associated with the products as conversion advanced. This
study provided novel insights into the heterogeneous conversion mechanisms of XPAHs on particulate
matter, offering valuable contributions to the understanding of their environmental behavior and impact.


**ASSOCIATED CONTENT**
**Author contributions**
YY, RJ and MZ conceived the study and wrote the paper, YY, YL, GZ, SZ and XL performed the
measure ments and collected data. All authors contributed to the data analysis and review of the paper.
**Competing Interests**
he authors declare that they have no conflict of interest.
**Data availability**
All data are available from the authors upon request by contacting Rong Jin (jinrong@ucas.ac.cn).
**Supporting information**
Including detailed information on chemicals, sample testing and analysis data, screening of
chemical properties and toxicity prediction information, relevant parameters and data from simulation



experiments, as well as chromatograms for product identification.

**Acknowledgement**

The authors acknowledge the financial support provided by the Natural Science Foundation of Zhejiang
Province (Grant No. LQ22B070009), the National Natural Science Foundation of China (Grant No.
22106030), "Pioneer" and "Leading Goose" R&D Program of Zhejiang (Grant No. 2023C03157) and the
Research Funds of Hangzhou Institute for Advanced Study, UCAS (Grant No. 2023HIAS-Y014,
2022ZZ01017).





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
