# Peer review of "Heterogeneous Phototransformation of Halogenated Polycyclic Aromatic"

_EGUsphere, 2024_

## Referee Comment (RC1)

**Reviewer's comments**

Comments on "Heterogeneous Phototransformation of Halogenated Polycyclic Aromatic Hydrocarbons: Influencing Factors, Mechanisms and Products" egusphere-2024-2814

This study provides a comprehensive investigation into the emerging persistent organic pollutants, Chlorinated and brominated polycyclic aromatic hydrocarbons (XPAHs). Concentration of XPAHs on atmospheric particulate matter were analyzed, and key influencing factors were explored by establishing relationships between concentration profiles and meteorological factors. Subsequently, through laboratory simulations, these factors were examined in depth, and the corresponding transformation mechanisms and products were identified. The overall idea of the study is complete and novel, and the experimental design is rigorous. The language and text are clear. However, there are still some issues that need to be considered and discussed. The comments are listed as follows.

**Specific comments:**

Line 27: "Transformation mechanism; Transformation pathways" or "Transformation pathways; Transformation mechanism", which sequence would be more appropriate?

Lines 122-123: How much flux does a xenon lamp have? Is this comparable to the radiation from the sun?

Lines 173-174: "The detection limits (LODs) ranged from 0.17 to 1.9 fg/m3 for ClPAHs, and from 0.23 to 1.6 fg/m3 for BrPAHs." Is there any reason for the variation of the values of LOD of ClPAHs and BrPAHs?

Lines 188-189: For example, "12.1±16.9 pg/m3" indicates the mean value, so it may not require to use mean with the mean value like (mean: $12.1 \pm 16.9$ pg/m³). So, it is suggested, no need to use mean with the mean value throughout the text,

Line 205: Kitazawa et al" not "KITAZAWA et al",

Line 206: "------------- highest molar mass ----------------", Not "------- height molecular weight-

------",

Lines 2016, 219 and other parts, please check, weather article "the" is required before "winter" or other seasons?

Lines 280-281: ---------- bromination degree ---------- or ----------------- degree of bromination ------------,

Lines 294-295: hydroxyl radical, •OH,

Line 302: ----------- which could participate------------,

Line 306: ------------ 30 °C--------------,

Lines: 311-314: The authors describe that the transformation rate of BrPAHs is higher than that of ClPAHs. This is due to the lower bond energy of C-Br (291 kJ/mol) compared to C-Cl (345 kJ/mol). Authors can think about the effect of humidity on Cl and Br atoms. The size of Cl atom is smaller that that of Br, so affinity of water molecule (vapor phase) would be higher for Br atom,

Line 433: The authors describe the effect of H2O2 as ROS species on the phototransformation, but what is the contribution of ozone on the phototransformation?.

---

## Referee Comment (RC2)

**EGUsphere Journal**

**Manuscript Reviewing Request**

*This bottom is my Official Review of the submission No. Egusphere-2024-2814*

**MS No.: egusphere-2024-2814**

**Title: Heterogeneous Phototransphormation of Halogenated Polycyclic Aromatic Hydrocarbons: Influencing Factors, Mechanisms and Products**

**Authors: Yueyao Yang et al.**

**General Comments**:

The paper presents the results of an investigation focused on particulate chlorinated and brominated PAHs affecting the atmosphere of Beijing, China RP, and on processes developing in the air, leading to their decomposition because of reaction with oxidants; the impact of meteo-climatic conditions is also elucidated, and the resulting effect on toxicity is evaluated. The investigation looks finely conceived and conducted, the database seems sufficient and suitable for applying the statistical approach adopted for assessing the role of forcing factors. The results seem of concern and suitable for reaching the research purposes. Thus, the subject enjoys enough concern to consider for publishing in EGUsphere.

The manuscript looks, in general, well written, and the results finely discussed. English language looks fluent, and just some points seem requiring to be improved, overall of editing nature. My general evaluation of the manuscript remains positive and an only minor revision seems necessary, according to remarks reported bottom. This further work of Authors would allow accepting the paper to publish in EGUsphere.

**Particular remarks**

Line 018: insert "that" after "found".
Line 019: insert "the contribution of" before "PM1".
Line 022: cut "transformation".
Line 026: change "XPAHs, with this process being" with "XPAHs; this process was".
Line 027: change "increased" with "high".
Line 029: change "We then" with "According to that, we".
Line 038: change "dechlorination" with "dehalogenation".
Line 040: insert "cumulatively," before "XPAHs".
Line 041: change "garnered" with "gained".
Line 052: insert "means of" before "detection".
Line 054: insert "i.e.," before "the environmental".
Line 058: change "atmospheric organic chemical species" with "organic substances affecting the atmosphere".
Line 061: change "study" with "studies".
*Line 99-100: see comment at Line 201.*
Line 130: change "was" with "were".
Line 181-182: change "652.7" with "653" and "167.6" with "167l".
*Line 201: Triph (i.e., triphenylene) was not listed among PAHs at lines 99-100. I suggest introducing it there.*

Line 208: I suppose Br2Byr is BrPyr.
Line 208: change "presented" with "represented".
Line 209: change "departure" with "difference" or "change".
Line 225: change "subject" with "subjected".
Line 256: change "transformated" with "transformed".

**Suggestions for the Editor**:
In conclusion, in my opinion the paper needs a minor revision before accepting to publish in EGUsphere.

Best regards,

---

## Author Response (AR2)

Prof. Allan Bertram

*Atmospheric Chemistry and Physics*

February 12, 2025

Dear Prof Bertram,

Thank you for organizing the review process for our manuscript entitled "***Heterogeneous Phototransformation of Halogenated Polycyclic Aromatic Hydrocarbons: Influencing Factors, Mechanisms and Products***" (Manuscript ID: Egusphere-2024-2814)**.**

We sincerely appreciate the editor and reviewers for their valuable and insightful comments. Detailed responses to the reviewers' feedback are provided below, and the manuscript has been revised accordingly to address and clarify the highlighted points.

Yours sincerely,

Rong Jin, Ph.D & Assoc. Prof.

Hangzhou Institute for Advanced Study, University of Chinese Academy of Sciences

E-mail: jinrong@ucas.ac.cn;

**Reviewer: Prof. Ahsan Habib**

This study provides a comprehensive investigation into the emerging persistent organic pollutants, Chlorinated and brominated polycyclic aromatic hydrocarbons (XPAHs). Concentration of XPAHs on atmospheric particulate matter were analyzed, and key influencing factors were explored by establishing relationships between concentration profiles and meteorological factors. Subsequently, through laboratory simulations, these factors were examined in depth, and the corresponding transformation mechanisms and products were identified. The overall idea of the study is complete and novel, and the experimental design is rigorous. The language and text are clear. However, there are still some issues that need to be considered and discussed. The comments are listed as follows.

**Response:** We sincerely appreciate your valuable feedback and are truly grateful for your thoughtful efforts. We have carefully considered all the comments provided, with our revisions highlighted in blue in the marked manuscript.

**Specific comments:**

**Comment 1:** Line 37: "Transformation mechanism; Transformation pathways" or "Transformation pathways; Transformation mechanism", which sequence would be more appropriate?

**Response 1:** We greatly appreciate your thoughtful suggestion. In our experiments, we firstly explored the influencing factors and general transformation mechanisms of XPAHs (Section 3.3, Line 251-285). Subsequently, we identified the transformation products of XPAHs, and clarified their specific transformation pathways (Section 3.5, Line 331- 397). The mechanism encompasses not only the influencing factors but also the pathways involved. Therefore, we prefer the title "Transformation mechanism; Transformation pathways" .

**Comment 2:** Lines 122-123: How much flux does a xenon lamp have? Is this comparable to the radiation from the sun?

**Response 2:** We utilized a xenon lamp with a light intensity of 100 mW/cm² (as mentioned in Line 123), and the typical luminous efficacy of the xenon lamp is 100 lm/W. Using the formula for light flux, $\Phi = I \times A \times \eta$, we calculated the light flux of the xenon lamp to be 100,000 lm per square meter.

Additionally, the light intensity of sunlight at noon at the sampling sites is also approximately 100 mW/cm$^2$ , and an average luminous efficacy ranging from 100 to 120 lm/W. Therefore, in terms of both light intensity and light flux, this study simulates sunlight illumination.

The revised text is shown as below.

Line 124-126:

In this study, we designed a photo-transformation device that provided complete confinement and precise control over the experimental conditions. The reaction unit employed a xenon lamp (light intensity: 100 mW·cm$^{-2}$, luminous efficacy: 100 lm/W) as the primary light source with an AM1.5 filter, which can achieve a good fit with the sunlight spectrum, effectively simulating the outdoor solar radiation (the average luminous efficacy ranged from 100 to 120 lm/W at noon at the sampling sites)

**Comment 3:** Lines 173-174: "The detection limits (LODs) ranged from 0.17 to 1.9 fg/m$^3$ for ClPAHs, and from 0.23 to 1.6 fg/m$^3$ for BrPAHs." Is there any reason for the variation of the values of LOD of ClPAHs and BrPAHs?

**Response 3**: The LODs for different ClPAH or BrPAH monomers vary due to differences in their monomeric chemical structures and properties. These differences can influence their response and stability during the analytical process. Additionally, different substituents may result in variations in signal intensity across certain analytical methods, thereby affecting the detection limits.

The revised text is inserted as below.

Line 177-180:

The LODs for ClPAH or BrPAH monomers vary due to differences in chemical structures, properties, and substituents, which influence their stability, response, and signal intensity during analysis, ultimately affecting detection limits.

**Comment 4:** Lines 188-189: For example, "12.1±16.9 pg/m3" indicates the mean value, so it may not require to use mean with the mean value like (mean: 12.1 ± 16.9 pg/m³). So, it is suggested, no need to use mean with the mean value throughout the text.

**Response 4:** We appreciate your valuable questions. We have made revisions regarding the issue of the average value in the article (Line 186-198, Page 8).

The revised text is shown as below.

Line 186-188:

During the sampling period, PM concentrations ranged from 41.5 to 653 µg/m3, with an average concentration of 130 µg/m$^3$ (Fig. S2a). PM1 had the highest proportion, with an average of 47.0%, while the proportions of PM$_{1-2.5}$ and PM$_{2.5-10}$ were comparable.

Line 193-198:

The concentrations of $\sum_{21}$ClPAHs in the particles ranged from 0.6 to 61.5 pg/m$^3$ (mean: 12.1 pg/m$^3$), while those of $\sum_{18}$BrPAHs ranged from N.D. to 5.4 pg/m$^3$ (mean: 0.6 pg/m$^3$) during the sampling period (Fig. 2a). These values are lower than those reported in previous studies, such as Beijing, China ($\sum_{19}$ClPAHs: 129 pg/m$^3$, $\sum_{19}$BrPAHs: 9.5 pg/m$^3$) (Jin et al. 2017a); Ulsan, South Korea ($\sum_{11}$BrPAHs: 1.62 pg/m$^3$) (Vuong et al. 2020); and Shizuoka, Japan ($\sum_{20}$ClPAHs: 133 pg/m$^3$) (Ohura et al. 2013).

**Comment 5:** Line 205: "Kitazawa et al" not "KITAZAWA et al".

Response 5: We greatly appreciate you bringing this issue to our attention. We apologize for the oversight in incorrectly spelling the name of this researcher, and we have made the necessary correction in the article (Line 210, Page 8).

The revised text is shown as below.

Line 210:

These distributions aligned with findings of prior studies (Jin et al. 2017a; Kitazawa et al. 2006; Ma et al. 2013).

**Comment 6:** Line 206: "------------- highest molar mass ----------------", Not "------- height molecular weight------------".

**Response 6:** Thank you for your insightful suggestion regarding the terminology. We change the "molecular weight" into "molar mass" in the revised manuscript.

The revised text is shown as below.

Line 211:

It is worth noting that 6-ClBaP, characterized by the highest molar mass and highest toxic equivalent factor investigated within ClPAHs investigated in our study

**Comment 7:** Lines 2016, 219 and other parts, please check, weather article "the" is required before

"winter" or other seasons?

**Response 7:** Thank you for your reminder. We have made revisions to the relevant content in the article (Line 221-230 9).

The average concentrations of PM and BrPAHs reached nadirs during summer and autumn while showing higher levels in the spring and winter (**Fig. S2d**). Conversely, ClPAH concentrations remained relatively stable during the spring and the summer, decreased in the autumn, and peaked in the winter. The seasonal characteristics of ClPAHs with different parent PAH structures also varied (**Fig. 2c**). Except for ClBaA, the remaining ClPAHs reached their highest concentrations during the winter. ClFluor showed elevated concentrations in both the spring and winter, whereas ClAnt demonstrated higher concentrations in the autumn and the winter. ClPhe maintained relatively consistent concentrations across the remaining three seasons. The seasonal characteristics of ClPAHs and BrPAHs (**Fig. 2d**) also differed. Concentrations of BrPAHs varied significantly with the seasons, with no congener detected in the summer and high concentrations of BrPhe, BrAnt, and BrPyr in the spring and winter, likely influenced by climatic conditions such as temperature and sunshine.

**Comment 8:** Lines 280-281: ---------- bromination degree ---------- or ----------------- degree of bromination----------.

**Response 8:** We have made the necessary revision in the article. Thank you for your understanding (Line 279-281, Page 13).

The revised text is shown as below.

Line 286:

Additionally, the increase in degree of bromination didn't appear to have a notable effect on the transformation rate of BrPAHs, which differed from ClPAHs.

**Comment 9:** Lines 294-295: hydroxyl radical, •OH.

**Response 9:** Thank you for your suggestion. We have corrected "OH radicals" to "hydroxyl radical" in the manuscript (Line 308, Page 14).

The revised text is shown as below.

Line 308:

Possible reasons could be that the addition of photo irradiation drove the formation of hydroxyl radical

(・OH), which could participate the breakdown of molecules (Zhang et al. 2023).

**Comment 10:** Line 302: ----------- which could participate -----------.

**Response 10**: Thank you for pointing out this issue in our manuscript. We have made the necessary modifications in the manuscript (Line 308, Page 14).

The revised text is shown as below.

Line 308:

Possible reasons could be that the addition of photo irradiation drove the formation of hydroxyl radical (・OH), which could participate in the breakdown of molecules (Zhang et al. 2023).

**Comment 11:** Line 306: ------------ 30 °C ------------.

Response 11: Thank you for your suggestion. We have checked the temperature units in the article and made the necessary revisions (Line 313, Page 14).

The revised text is shown as below.

Line 313:

Transformation ratios of XPAHs increased as temperatures rose, with the most significant transformation observed at 30 °C (Fig. 4b), indicating that the elevated temperature promote the transformation of XPAHs.

**Comment 12:** Lines: 311-314: The authors describe that the transformation rate of BrPAHs is higher than that of ClPAHs. This is due to the lower bond energy of C-Br (291 kJ/mol) compared to C-Cl (345 kJ/mol). Authors can think about the effect of humidity on Cl and Br atoms. The size of Cl atom is smaller that of Br, so affinity of water molecule (vapor phase) would be higher for Br atom.

**Response 12:** Thank you for your suggestion. We indeed overlooked the influence of atomic size. The radius of the chlorine atom (Cl) is smaller than that of the bromine atom (Br), which means that, under the same conditions, the interaction between water molecules and bromine atoms may be stronger. We have added this information to the article (Line 319-322, Page 14).

The revised text is shown as below.

In both photo irradiation and dark conditions, the transformation ratios of BrPAHs (Fig. S7c and Fig. S7d) exceeded those of ClPAHs. This phenomenon can be attributed to the differences in bond strength

(the bond energy of C-Br is lower at 291 kJ/mol, whereas the bond energy of C-Cl is 345 kJ/mol) (Ohura et al. 2009) and atomic size (the radius of the Cl atom is approximately 99 pm, while the radius of the Br atom is around 114 pm) (Shannon 1976).

**Comment 13:** Line 433: The authors describe the effect of H2O2 as ROS species on the phototransformation, but what is the contribution of ozone on the phototransformation?

**Response 13:** Thank you for your suggestion. $H_2O_2$ is one of the representative ROS species. We chose $H_2O_2$ as a variable because its liquid form allows for easier control of concentration, thereby enhancing the precision of our experiments. Additionally, both $H_2O_2$ and $O_3$ decompose into singlet oxygen, which can oxidize XPAHs and promote their decomposition. Therefore, the oxidation processes of $H_2O_2$ and other ROS could be comparable to some extent.

The following sentence has been inserted in the revised manuscript:

Line 139:

H2O2 was selected for its representativeness among reactive oxygen species and ease of control.

**Reviewer: Anonymous Referee #2**

The paper presents the results of an investigation focused on particulate chlorinated and brominated PAHs affecting the atmosphere of Beijing, China RP, and on processes developing in the air, leading to their decomposition because of reaction with oxidants; the impact of meteo-climatic conditions is also elucidated, and the resulting effect on toxicity is evaluated. The investigation looks finely conceived and conducted, the database seems sufficient and suitable for applying the statistical approach adopted for assessing the role of forcing factors. The results seem of concern and suitable for reaching the research purposes. Thus, the subject enjoys enough concern to consider for publishing in EGUsphere.

The manuscript looks, in general, well written, and the results finely discussed. English language looks fluent, and just some points seem requiring to be improved, overall of editing nature. My general evaluation of the manuscript remains positive and an only minor revision seems necessary, according to remarks reported bottom. This further work of Authors would allow accepting the paper to publish in EGUsphere.

**Response:** We sincerely appreciate your valuable feedback and are truly grateful for your thoughtful contributions. We have carefully considered all of the comments provided, and our revisions are highlighted in blue in the marked manuscript.

**Comment 1:** Line 018: insert "that" after "found".

**Response 1**: Thank you for your suggestion. The changes have been made in the manuscript (Line 18, Page 2.

The revised text is shown as below.

Line 17-19:

In this study, we collected PM samples of varying sizes over a year for XPAH analysis and found that the average concentrations of XPAHs peaked in the winter and were dominated by the contribution of PM1 (47.0%).

**Comment 2:** Line 019: insert "the contribution of" before "PM1".

**Response 2:** Thank you for your feedback. The changes have been made in the manuscript (Line 19,

Page 2).

The revised text is shown as below.

Line 17-19:

In this study, we collected PM samples of varying sizes over a year for XPAH analysis and found that the average concentrations of XPAHs peaked in the winter and were dominated by the contribution of PM1 (47.0%).

**Comment 3:** Line 022: cut "transformation".

**Response 3:** Thank you for your feedback. The revisions have been made in the manuscript (Line 22, Page 2).

The revised text is shown as below.

Line 21-22:

Hence, controlled laboratory experiments were conducted to explore the influence of particle size, sunlight duration, temperature, humidity, and oxidant concentrations on XPAHs transformation.

**Comment 4:** Line 026: change "XPAHs, with this process being" with "XPAHs; this process was".

**Response 4**: Thank you for your feedback. The revisions have been made in the manuscript (Line 26, Page 2).

The revised text is shown as below.

Line 25-28:

Furthermore, the photo irradiation promoted the heterogeneous transformation of XPAHs; this process was accelerated by the increased concentrations of reactive oxygen species and elevated temperature, peaking at the humidity level of 45%.

**Comment 5:** Line 027: change "increased" with "high".

**Response 5**: Thank you for your feedback. However, I believe "increased" is more appropriate than "high" in the manuscript, as the dynamic processes of increasing reactive oxygen species concentration and rising temperature accelerate the heterogeneous transformation.

**Comment 6:** Line 029: change "We then" with "According to that, we".

Response 6: Thank you for your suggestion. The revisions have been made in the manuscript (Line 28-29, Page 2), and the logic is now clearer.

The revised text is shown as below.

Line 28-30:

According to that, we then proposed phototransformation pathways for XPAHs, suggesting a mechanism involving dehalogenation followed by oxidation.

**Comment 7:** Line 030: change "dichlorination" with "dehalogenation".

**Response 7:** Thank you for your suggestion. The revisions have been made in the manuscript (Line 30, Page 2). It was our oversight, and the changes make it more appropriate.

The revised text is shown as below.

Line 28-30:

According to that, we then proposed phototransformation pathways for XPAHs, suggesting a mechanism involving dehalogenation followed by oxidation.

**Comment 8:** Line 040: insert "cumulatively," before "XPAHs".

**Response 8:** Thank you for your suggestion. The revisions have been made in the manuscript (Line 40, Page 3).

The revised text is shown as below.

Line 40-44:

Chlorinated and brominated polycyclic aromatic hydrocarbons (ClPAHs and BrPAHs), cumulatively referred to as XPAHs, are halogenated derivatives of polycyclic aromatic hydrocarbons (PAHs) that have gained considerable attention in recent years due to their heightened persistence, toxicity, and bioaccumulation relative to their parent PAHs.(Jin et al. 2017b; Ma et al. 2013; Nishimura et al. 2017; Ohura et al. 2013)

**Comment 9:** Line 041: change "garnered" with "gained".

**Response 9:** Thank you for your suggestion. The revisions have been made in the manuscript (Line 41, Page 3).

The revised text is shown as below.

Line 40-43:

Chlorinated and brominated polycyclic aromatic hydrocarbons (ClPAHs and BrPAHs), collectively referred to as XPAHs, are halogenated derivatives of polycyclic aromatic hydrocarbons (PAHs) that have gained considerable attention in recent years due to their heightened persistence, toxicity, and bioaccumulation relative to their parent PAHs.(Jin et al. 2017b; Ma et al. 2013; Nishimura et al. 2017; Ohura et al. 2013)

**Comment 10:** Line 052: insert "means of" before "detection".

**Response 10:** Thank you for your suggestion. The revisions have been made in the manuscript (Line 52, Page 3).

The revised text is shown as below.

Line 50-53:

The sources, such as industrial thermal processes (Yang et al. 2022b), electronic waste decomposition (Wang et al. 2022), and vehicular emissions (Deng et al. 2023) have been identified, by means of detection of XPAHs in the stack gas and fly ash emitted from these sources (Jin et al. 2017c; Nishimura et al. 2017; Yang et al. 2022a).

**Comment 11:** Line 054: insert "i.e.," before "the environmental".

**Response 11**: Thank you for your suggestion. The revisions have been made in the manuscript (Line 54, Page 3), and the logic is now more rigorous.

The revised text is shown as below.

Line 53-54:

However, a significant aspect of research appears to have been overlooked: i.e. the environmental transformation.

**Comment 12:** Line 058: change "atmospheric organic chemical species" with "organic substances affecting the atmosphere".

Response 12: Thank you for your suggestion. The revisions have been made in the manuscript (Line 58, Page 3).

The revised text is shown as below.

Line 57-58:

The photochemical processes have been verified to represent a marked elimination pathway for organic substances affecting the atmosphere. (Hu et al. 2021; Laskin et al. 2015; Malecha and Nizkorodov 2016)

**Comment 13:** Line 061: change "study" with "studies".

Response 13: Thank you for your suggestion. The revisions have been made in the manuscript (Line 62, Page 3).

The revised text is shown as below.

Line 62-64:

Studies by Ohura et al. also confirmed the photochemical transformation of ClPAHs when being exposed to light in cyclohexane solvent (Ohura et al. 2008).

**Comment 14:** Line 99-100: see comment at Line 201.

**Response 14:** Thank you for your suggestion. The revisions have been made in the manuscript (Line 99, Page 5).

The revised text is shown as below.

Line 98-100:

The parent PAHs are abbreviated as follows: Nap (naphthalene), Phe (phenanthrene), Ant (anthracene), Triph (triphenylene), Fluor (fluoranthene), Pyr (pyrene), BaA (benz[a]anthracene), and BaP (benzo[a]pyrene).

**Comment 15:** Line 130: change "was" with "were".

**Response 15:** Thank you for your suggestion. However, we believe that "was" is more appropriate than "were" in the manuscript because "20 μL of the XPAHs mixture (1 mg/L) or individual congener solution" represents a single system and should be expressed in singular form.

**Comment 16:** Line 181-182: change "652.7" with "653" and "167.6" with "167".

Response 16: Thank you for your suggestion. However, we have revised these across the manuscript.

Line187-188:

During the sampling period, PM concentrations ranged from 41.5 to 653 μg/m3, with an average

concentration of 130 μg/m3 (Fig. S2a).

**Comment 17:** Line 201: Triph (i.e., triphenylene) was not listed among PAHs at lines 99-100. I suggest introducing it there.

**Response 17:** Thank you for your suggestion. This information has been added in lines 99–100, as stated in Response 14.

**Comment 18:** Line 208: I suppose Br2Byr is Br2Pyr.

**Response 18:** Thank you for your suggestion. The revisions have been made in the manuscript (Line 215, Page 8), and we will strive to minimize similar minor errors in the future.

The revised text is shown as below.

Line 215:

BrPAHs are predominantly constituted by 2-BrPhe, 9-BrPhe, 7-BrBaA, and 1,6-Br2Pyr.

**Comment 19:** Line 208: change "presented" with "represented".

**Response 19:** Thank you for your feedback. The revisions have been made in the manuscript (Line 207, Page 8), and using "represented" is indeed more appropriate.

The revised text is shown as below.

Line 216:

This represented a notable departure from previous literature which predominantly identified 3-BrFluor, 1,8-Br2Ant, and 1-BrPyr as the primary congeners of BrPAHs in atmospheric PM in Beijing. (Jin et al. 2017a)

**Comment 20:** Line 209: change "departure" with "difference" or "change".

Response 20: Thank you for your feedback. The revisions have been made in the manuscript (Line 208, Page 8), and using "difference" is indeed more appropriate.

The revised text is shown as below.

Line 215:

This represented a notable difference from previous literature which predominantly identified 3-BrFluor, 1,8-Br2Ant, and 1-BrPyr as the primary congeners of BrPAHs in atmospheric PM in Beijing. (Jin et al. 2017a)

**Comment 21:** Line 225: change "subject" with "subjected".

Response 21: Thank you for your feedback. The revisions have been made in the manuscript (Line 225, Page 9), and using "subjected" is indeed more appropriate.

The revised text is shown as below.

Line 233:

This disparity can be attributed to the ease of generation from sources and greater atmospheric stability of ClPAHs, while BrPAHs may be subjected to influences from atmospheric processes (Ohura et al. 2009).

**Comment 22:** Line 256: change "transformated" with "transformed".

Response 22: Thank you for your feedback. The revisions have been made in the manuscript (Line 255, Page 12), and using "transformed" is indeed more appropriate.

The revised text is shown as below.

Line 264:

The transformation ratios of XPAHs were calculated based on the ratios of transformed XPAHs to the initial concentration ((C0-Ct)/C0). With the increase of irradiation time, a general trend of transformation was observed across XPAH congeners (Fig. 3).